# Correlation between the Chemical Structure of (Meth)Acrylic Monomers and the Properties of Powder Clear Coatings Based on the Polyacrylate Resins

**DOI:** 10.3390/ma17071655

**Published:** 2024-04-03

**Authors:** Katarzyna Pojnar, Barbara Pilch-Pitera

**Affiliations:** 1Doctoral School of Engineering and Technical Sciences, Rzeszow University of Technology, ul. Powstańców Warszawy 12, 35-959 Rzeszów, Poland; d521@stud.prz.edu.pl; 2Department of Polymers and Biopolymers, Faculty of Chemistry, Rzeszow University of Technology, ul. Powstańców Warszawy 6, 35-959 Rzeszów, Poland

**Keywords:** polyacrylate resin, powder coating properties, low-temperature coatings

## Abstract

This paper presents studies on the influence of the chemical structure of (meth)acrylic monomers on the properties of powder coatings based on polyacrylate resins. For this purpose, a wide range of monomers were selected—2-hydroxyethyl methacrylate (HEMA), methyl methacrylate (MMA), *n*-butyl acrylate (*n*BA), *tert*-butyl acrylate (*t*BA), dodecyl acrylate (DA), ethyl acrylate (EA) and benzyl acrylate (BAZ)—for the synthesis of the polyacrylate resin. The average molecular mass and molecular mass distribution of the synthesized resins were measured by gel permeation chromatography (GPC). The glass transition temperature (T*_g_*) and viscosity of polyacrylate resins were determined by using differential scanning calorimetry (DSC) and a Brookfield viscometer. These parameters were necessary to obtain information about storage stability and behavior during the application of powder clear coatings. Additionally, DSC was also used to checked the course of the low-temperature curing reaction between the hydroxyl group contained in the polyacrylate resin and the blocked polyisocyanate group derived from a commercial agent such as Vestanat B 1358/100. The properties of the cured powder clear coatings were tested, such as: roughness, gloss, adhesion to the steel surface, hardness, cupping, scratch resistance, impact resistance and water contact angle. The best powder clear coating based on the polyacrylate resin L_HEMA/6MMA/0.5*n*BA/0.5DA was characterized as having good scratch resistance (550 g) and adhesion to the steel surface, a high water contact angle (93.53 deg.) and excellent cupping (13.38 mm). Moreover, its crosslinking density (CD) and its thermal stability was checked by using dynamic mechanical analysis (DMA) and thermogravimetric analysis (TGA).

## 1. Introduction

Currently, specific customer requirements and EU directives are increasing, necessitating the development of new products that meet the new requirements set for the paints and coatings market [1,2,3]. The rising concerns over energy consumption, high emissions, harmful and toxic gases, liquid coatings containing large amounts of volatile organic compounds (VOCs), the utilization of petroleum-based raw materials in large quantities and other significant issues, highlight the need for new approaches. Powder coatings, in comparison to liquid paints, do not contain VOCs, and their application meets requirements such as high economy, efficiency, sustainability and quality (for example: the new powder coatings line developed by Ecoline company, Poland) [4,5]. Depending on the raw materials used for the powder coating composition, it is possible to obtain a finished product cured at low temperature or UV cured, thereby reducing production costs and energy consumption [6]. Moreover, many researchers have conducted studies on the use of biobased or renewable products. Li et al. described biobased UV-curable (powder) coating resins based on limone-derived polycarbonates obtained from orange oils and carbon dioxide [7]. The UV-cured powder coatings used limone-derived polycarbonates as binders characterized high transparency, good acetone resistance, high pencil hardness (H–2 H) and high König hardness (174–199 s).

The most common resins used for powder coatings are based on epoxy and polyester [8,9,10]. Due to compatibility issues with other resins such as acrylic, polyacrylic resins are less frequently used. Separate production lines are required to produce powder coatings based on polyacrylate resins. Despite this disadvantage, acrylic powder coatings are characterized by high physical and mechanical properties, very good gloss and color durability, making them ideal for outdoor use compared to polyesters or epoxy resin. These properties result from the chemical structure of (meth)acrylic monomers. Polyacrylic resin can contain reactive hydroxyl-, carboxyl- or glycidyl groups [11]. Polyacrylates containing hydroxyl groups exhibit excellent weather durability, but their drawback is limited impact resistance. However, the final properties of the coating are also influenced by the appropriate crosslinking agent. A commonly use curing agent, hydroxyl functional acrylic resin, is a blocked polyisocyanate (PIC). Powder coating based on acrylic-resin-containing hydroxyl groups can cure below 160 °C creating low-temperature systems [12]. Acrylic resins containing glycidyl groups offer several benefits, including effective corrosion protection, resilience against weathering and high adhesion [13]. Moreover, epoxy functional acrylic resin can be crosslinked by UV radiation in the presence of cationic photoinitiators [14]. The possibility of UV curing in a shorter time, like low-temperature systems, allows the application of powder coatings to heat-sensitive surfaces such as wood, MDF board or even polymer composite [15]. Okada et al. described an acrylic/polyester hybrid powder coating. These coatings are characterized by good appearance and mechanical properties such as toughness and flexibility. However, a notable drawback of these coatings is their insufficient resistance to weathering [16]. Generally, acrylic monomers, thanks to the reactive double bonds, offer a wide range of opportunities. It is also possible to use acrylic monomers for Atom Transfer Radical Polymerization (ATRP) or even for the production of self-healing coatings, but everything depends on the appropriative selection of the monomers used [17].

The primary issue addressed in this article is the weak flexibility of acrylic resin. However, the selection of raw materials with an appropriate structure may alter the interactions between the polymer chains and allow a compromise between these properties to be achieved. There are studies on this topic in the scientific literature. Shi et al. synthesized hyperbranched polyurethane acrylate, which was blended with epoxy acrylate and tripropylene glycol diacrylate (TPGDA) to prepare UV-cured films [18]. They found that the incorporation of hyperbranched polyurethane acrylate could enhance the mechanical strength to 19 MPa, elongation at break to 310% and impact resistance to 26.9 MJ m^−3^ of UV-cured films simultaneously. However, the synthesis of hyperbranched polyurethane acrylate often requires complex and sophisticated synthesis routes [19]. Coatings with low cupping tend to crack faster. The brittleness of the coating may cause damage, leading to corrosion and lack of tightness. Li et al. described a flexible and self-healing epoxy acrylic coating by introducing dynamic disulfide bonds and dangling chains into the resin structure. Furthermore, nano-SiO_2_ was sprayed to construct the microsurface structure to endow the coating super-hydrophobicity [20].

Obtaining innovative, specific acrylic resins can significantly expand their applications and meet the requirements set by customers and legal regulations. Nevertheless, formulating a material that can exhibit all desired properties and functions, especially if they are opposing (e.g., flexibility and hardness), remains challenging. Therefore, further research in this area needed, especially since few studies on acrylic resins for powder coating have been published in the scientific literature.

Therefore, the aim of this work is to investigate the correlation between the chemical structure of appropriate (meth)acrylic monomers and the properties of powder clear coatings. Gel permeation chromatography (GPC) and differential scanning calorimetry (DSC) were used to determine the molar mass distribution and the glass transition temperature T*_g_*. Moreover, the viscosity was measured to evaluate the storage stability and behavior during the application of powder coatings. Nuclear magnetic resonance (NMR) was used to confirm the chemical structure of the resins.

The visual and mechanical properties of cured powder coatings were investigated using tests such as roughness, gloss, adhesion to the steel surface, hardness, cupping, scratch resistance, impact resistance and water contact angle. The crosslinking density and thermal stability were checked using dynamic mechanical analysis (DMA) and thermogravimetric analysis (TGA). Based on the conducted research, coatings with optimal properties were selected and correlated with the structure of monomers used for the synthesis of the resin.

## 2. Experimental Section

### 2.1. Materials

Materials and reagents used for the synthesis of polyacrylate resins:2-hydroxyethyl methacrylate (HEMA) (Sigma Aldrich, Darmstadt, Germany);Methyl methacrylate (MMA) (Sigma Aldrich, Darmstadt, Germany);*n*-butyl acrylate (*n*BA) (Sigma Aldrich, Darmstadt, Germany);*tert*-butyl acrylate (*t*BA) (Sigma Aldrich, Darmstadt, Germany);Dodecyl acrylate (DA) (Sigma Aldrich, Darmstadt, Germany);Ethyl acrylate (EA) (Sigma Aldrich, Darmstadt, Germany);Ethyl methacrylate (EMA) (Sigma Aldrich, Darmstadt, Germany);Benzyl acrylate (BAZ) (Sigma Aldrich, Darmstadt, Germany);Free radical initiator of polymerization: azobisisobutyronitrile (AIBN) (Sigma Aldrich, Darmstadt, Germany).

Raw materials used for the powder clear coating preparation:Vestanat^®^B 1358/100 (Evonic Degussa, Marl, Germany);Degassing agent: benzoin (Sigma Aldrich, Darmstadt, Germany);Flow control agent: Byk 368P (Byk-Chemie, Wesel, Germany).

### 2.2. Synthesis of Polyacrylate Resins

For each synthesis, methyl methacrylate and 2-hydroxyethyl methacrylate were used. Depending on the type of resins, an appropriate amount of (meth)acrylate monomers were added (Table 1).

As an initiator of free radical polymerization in bulk of acrylate monomers, 1.7% azabisisobutyronitrile (AIBN) was used. All compounds were placed in a three-necked flask equipped with a reflux condenser, a thermometer, a magnetic stirrer and a nitrogen inlet (the reaction was sensitive to oxygen inhibition). The reaction mixture was heated at a temperature of 80 °C. Polymerization reaction commenced when the viscosity experienced a rapid increase. Subsequently, the non-solidified resin mixture was poured onto a PTFE mold for the purpose of solidification. To complete the polymerization process, the PTFE mold containing the resin was sealed and positioned in an oven at 80 °C for one hour. Following this, the mold was allowed to cool and the solidified resin was subjected to grinding. The synthesized resins were designated based on the molar ratio and names of the monomers utilized; for instance, HEMA/6MMA/*n*BA denotes a resin synthesized from the HEMA, MMA and *n*BA monomers in a molar ratio of 1:6:1.

### 2.3. Preparation of Powder Clear Coatings Based on Synthesized of Polyacrylate Resins

For the preparation of powder coatings, the following materials were used: Vestanat B 1358/100 (oxime blocked polyisocyanate, based on isophorone diisocyanate), 1% of benzoin and 2% of Byk 368P. The amount of Vestanat B 1358/100 was calculated based on the hydroxyl number of the acrylic resin (L_OH_ = 40 mgKOH/g), so that the molar ratio of the -NCO groups to the -OH groups was 1:1. The prepared composition was mixed, ground and extruded in a co-rotating twin-screw mini-extruder EHP 2 × 12 Sline from Zamak (Cracow, Poland). The temperature in the extruder was maintained as follows: zone I-80 °C, zone II-90 °C, zone III-100 °C and adapter-110 °C. The rotational speed of screw was set to 150 rpm. In the next step, the extrudate was cooled, pulverized and sieved in a 100 μm sieve. The final powder coatings were applied to the metal plates. Prior to application, the metal plates were cleaned in acetone, degreased, and passivated using zirconium phosphate conversion ESKAPHOR Z 2000C (Haug Chemie, Sinsheim, Germany). The powder coatings were applied by electrostatic gun PEM X-1 controlled by EPG Sprint X (CORONA) from Wagner (Wagner, Altstätten, Switzerland). The coatings were curing at 160 °C for 15 min. The obtained powder coatings were named according to the names of the resin used, e.g., L_HEMA/6MMA/*n*BA means a coating made from the resin HEMA/6MMA/*n*BA.

## 3. Measurements

### 3.1. Gel Permeation Chromatography (GPC)

Gel permeation chromatography (GPC) (ViscoTec, Töging a. Inn, Germany) was conducted utilizing RI detector Shodex RI-71, a Shimadzu LC-20AD isocratic pump, a ViscoTec degassing system, a PSS SDV guard column, and PSS SDV 100, 1000 and 10,000 Å columns with a grain diameter of 5 μm, packed with styrene divinylbenzene-type gel were used. All samples were dissolved in tetrahydrofuran (HPLC grade) containing 5 mmol/L of the resin at a temperature of 22 °C. The analysis was carried out by interpreting results based on the conventional calibration of columns with polystyrene standards. Prior to analysis, solutions were shaken for 24 h at ambient temperature and subsequently filtered through syringe filters with a diaphragm (PTFE) of 0.25 μm.

### 3.2. Viscosity

The viscosity was measured at 140 °C using a cone-plate CAP 2000+ viscometer (AMETEK Brookfield, Middleboro, MA, USA) equipped with a cone no. 6, according to PN-EN ISO 2884-1.

### 3.3. Differential Scanning Calorimetry (DSC)

Thermal properties of resins and powder compositions were examined using a Mettler Toledo 822e calorimeter (Mettler Toledo, Columbus, OH, USA) equipped with Stare System software. The calibration of the DSC apparatus (Mettler Toledo r, Columbus, OH, USA) was conducted using indium and zinc standards supplied by Mettler Toledo (Columbus, OH, USA). Accurate weighing of sample (10 mg) was performed with a precision of 0.00001 g. Subsequently, the samples were hermetically sealed in standard 40 μL aluminum crucibles and positioned in the measuring chamber. The measurements were conducted in the temperature range from 0 to 200 °C, with a heating rate of 20 K min^−1^ and maintaining an atmosphere of nitrogen with a flow rate of 60 mL/min. This experimental setup facilitated the comprehensive investigation of the thermal behavior of the resin and powder compositions.

### 3.4. ^1^H-NMR Spectroscopy

The nuclear magnetic resonance (^1^H-NMR) spectra were acquired using a Bruker Avance II 500 MHz (Bruker BioSpin, Rheinstetten, Germany). This device featured a 5 mm nitrogen-cooled dual (BB-1H) cryoprobe. Tetramethylsilane served as the standard for internal referencing and chemical shift values were reported in parts per million (ppm). Deuterated chloroform (CDCl_3_) was used as a solvent. NMR Topspin 2.1 pl 8 software (Topspin 2.1 pl 8, Bruker BioSpin, Rheinstetten, Germany) was used.

### 3.5. Polymerization Test

Polymerization test was conducted in accordance with the technical requirements of the QUALICOAT [21]. The assessment of the sample took place after subjecting the coatings to rubbing with a swab soaked in methyl ethyl ketone (MEK). The swab was moved lightly back and forth 30 times in each direction. The classification of the sample was based on following criteria:The coating is matt and soft.The coating is matt and can be scratched with a nail.Slight loss of gloss.No noticeable changes.The polymerization test was performed twice for each coating.

### 3.6. Flowability

In accordance with PN-EN ISO 8130-11, the evaluation of paint flow from an inclined surface was carried out [22]. Cavities with a depth of 6.5 mm and a diameter of 25 mm were stamped into the steel test plates. Then, a paint sample of 0.5 g was placed in the cavity of the plate and positioned at 60° from horizontal in an oven heated to 160 °C for 15 min. After this time, the distance between the bottom edge of the cavity and the furthest point where the melted powder paint flowed was determined.

### 3.7. Thickness and Gloss

The gloss and thickness test were conducted using a gloss meter, specifically the micro-Tri-gloss-μ with a thickness measurement function, manufactured by BYK-Gardner (Geretsried, Germany). The assessment followed to the standards outlined in PN-EN ISO 2813 for gloss and PN-EN ISO 2808 for thickness [23,24]. Gloss was evaluated by measuring the intensity of light reflected from the coating at an incident angle of 60°. The thickness analysis was carried out using the same device, equipped with a built-in Fe/NFe sensor. Results for gloss and thickness were obtained by averaging measurements from ten trials for each sample.

### 3.8. Roughness

Measurements were carried out at a LT = 5600 mm and LC = 0.800 × 5N by the use of a Mar SurfPS1 apparatus from Mahr GmbH (Göttingen, Germany.), according to the PN-EN ISO 12085 standard [25]. The test focused on two roughness parameters, R*_a_* and R*_z_*. R*_a_* represents the arithmetic mean deviation from the baseline expressed in micrometers, while R*_z_* signifies the arithmetic mean of the five highest profile peaks decreased by the arithmetic mean of the five lowest profile depths. Roughness assessments were performed at ten different locations on the surface of the same coating and the average of ten measurements was taken as the final result.

### 3.9. Adhesion to the Steel Surface

The adhesion-to-the-steel-substrate test was conducted following the PN-EN ISO 2409 standard, utilizing the cross-cut method [26]. A specialized multi-cut tool equipped with six cutters, manufactured by Byk Gardner (Geretsried, Germany), was employed for this purpose. The cuts were made perpendicular to each other, forming a grid of squares with a 2 mm spacing. After the cutting process, any dust generated was brushed off the coating surface using a brush. Subsequently, a 50 mm wide adhesive tape with standardized peel force was applied to the surface. Upon breaking the tape, the appearance of the grid was assessed. The surface of the coatings was visually examined and classified on a 0–5 six-point scale, where 0 indicated no traces other than knife marks, and 5 represented almost complete or complete detachment of the coating. The test was repeated twice for the same coatings to ensure consistency in the results.

### 3.10. Hardness

Measurements of the coating hardness were carried out according to the standard PN-EN ISO 1522 using König Pendulum manufactured by BYK-Gardner (Geretsried, Germany) [27]. The relative hardness was calculated by dividing the arithmetic mean of the number of pendulum oscillations for the tested sample by glass constant, which is 171 pendulum oscillations. Three measurements were made for the same coating.

### 3.11. Scratch Resistance

The scratch resistance test was executed following the PN-EN ISO 1518 standard [28]. The cured coating was positioned on a test table that moved with increasing loads on the needle until the coating experienced a scratch. After the scratch occurred, the value of the needle load was recorded. The measurements were repeated four times using the same composition and the final result was determined based on the repeated measurements.

### 3.12. Water Contact Angle (WCA)

The water contact angle (WCA) was determined following the standard PN-EN ISO 19403-6:2020-08, employing an optical goniometer OCA15 EC from DataPhysics (Filderstadt, Germany) [29]. Drop contours were measured using the SCA20U computer software (Ravelin, London, United Kingdom) and the contact angle was subsequently calculated. Measurements were conducted at ten different locations on the surface of the same coating to capture variability. The final result was determined by averaging of these measurements to ensure a comprehensive representation of the coating’s water contact angle.

### 3.13. Impact Resistance

To determine the resistance of coatings to impacts, the SP1890 tester by TQC (Capelle aan den IJssel, Netherlands) was used in accordance with the PN-EN ISO 6272-1 standard [30]. The device consists of a tube with a marked height scale, a weight dropping from a specified height according to the standard onto a matrix where the test sample was placed. The test involved deformation of a thin sheet metal, on which the coating was applied, by the falling weight. Impact resistance was measured by assessing cracking or detachment of the deformed coating.

### 3.14. Cupping

The results were acquired through a manual SP4300 tester by TQC, following the PN-EN ISO 1520 standard [31]. The test involved pressing a spherical drawing punch into a clamped sheet until a crack appeared. The result was read at the point of coating cracking. To assess the repeatability of the results, three measurements were conducted on the same cured coating.

### 3.15. Dynamic Mechanical Analysis (DMA)

Dynamic mechanical analysis (DMA) measurements were conducted using DMA/SDTA861e unit from Mettler Toledo (Mettler-Toledo, Columbus, OH, USA). The analysis was performed in compressing mode with a constant frequency of 1 Hz, spanning a temperature range of 0–200 °C (heating rate 3 °C/min). The force amplitude was maintained at a maximum of 0.4 N and displacement amplitude at maximum 2.5 µm. The samples, with a diameter of approximately 15 mm, thickness 2.5 mm, geometry factor 14,15 1/m and weight 0.5 g were formed using a SPECA hydraulic press.

### 3.16. Thermogravimetric Analysis (TGA)

Thermogravimetric analysis was performed using Mettler Toledo TGA/DSC1 instrument (Mettler-Toledo, Columbus, OH, USA). The analyses were conducted in nitrogen in the temperature range of 25 to 700 °C, with a heating rate of 10 °C/min. The measurement conditions were as follows: sample weight ~5 mg, gas flow rate 50 cm^3^/min and 150 μL open alumina pan.

## 4. Results and Discussion

### 4.1. Choice of (Meth)Acrylic Monomers

The chemical structure of (meth)acrylate monomers plays a crucial role in determining the properties of resins and their suitability for various coating applications. The structure of these monomers, along with their molar ratio, affects factors such as thermal stability, rheology and other characteristics of the final product. With a wide variety of monomers available, there exists an extensive range of possible polyacrylate resin compositions. Hence, it is essential to comprehend how the chemical structure of (meth)acrylic monomers influences the resulting properties. Table 2 illustrates the impact of selected monomers on the resin performance.

The most commonly used monomer is methyl methacrylate (MMA) [32]. The presence of the -CH_3_ groups in the α-position of vinyl groups increases stiffness and improves the physical and mechanical properties of the resin [33]. For this reason, the second monomer often used for coating applications is n-butyl acrylate (*n*BA). The four-carbon aliphatic chain increases flexibility [34]. This increased flexibility would allow the application of this polymer on surfaces of various shapes, such as roofs or elements for agricultural and automotive equipment.

The use of the butyl acrylate isomer in the form of tert-butyl acrylate (*t*BA) also alters resin properties. The packed, tetrahedral structure of tert-butyl substituents promotes greater chain mobility, which increases hydrophobicity, reduces the glass transition temperature (T*_g_*) and maintains brilliant color [35,36]. Ethyl methacrylate (EMA) is often employed in the production of plastics with better resistance to weather conditions, while ethyl acrylate (EA) is more frequently used in various sectors of the chemical industry [37,38].

In addition to monomers containing aliphatic substituents in the ester group, a monomer containing an aromatic ring was also selected. Therefore, benzyl acrylate (BAZ) was chosen. The presence of the aromatic ring enhances mechanical strength and hardness. However, a disadvantage of using this monomer is the yellowing of coatings, which can lead to a decrease in adhesive and mechanical properties [39].

To optimize mechanical properties, dodecyl acrylate is an often-used monomer in acrylic resins. Due to the presence of long alkyl chain in ester groups, DA improves the tensile strength of polymers [40]. Additionally, the nonpolar long chain increases the hydrophobic properties [41].

Furthermore, monomers containing functional groups such as hydroxyl, carboxyl or epoxy groups are needed to ensure crosslinking ability. The 2-hydroxyethyl methacrylate was used in this work. The presence of polar hydroxyl groups increases hydrophilic properties, allowing for the creation of amphiphilic polymers [42]. Depending on the chemical structure of these monomers, properties related to elasticity or hardness can be controlled, and the appropriate curing agent can be selected to enable the crosslinking process [43,44].

### 4.2. Characterization of poly(meth)acrylate Resins

To characterize the obtained acrylic resins, gel permeation chromatography (GPC) and nuclear magnetic resonance (1H-NMR) spectra as well as viscosity and T*_g_* measurements were performed. Gel permeation chromatography (GPC) was used to determine the average molecular masses of acrylic resins (Table 3). The average molecular mass number of obtained resins ranged from 6 670 to 9 820 Da. The resin HEMA/6MMA/DA was characterized by the lowest number average molecular mass, while the resin containing *t*BA exhibited the highest number average molecular mass. These values of molecular mass depend on the reactivity of individual monomers, with *t*BA having higher reactivity than DA.

The high molecular weight of the resin used for powder coatings is desired to obtained good properties of the finished product [45]. However, excessively high molecular weight can lead to problems with the preparation of powder coatings due to excessively high viscosity. Conversely, a resin with too low a molecular mass may exhibit too low a viscosity and T*_g_*, resulting in the powder grains sticking together during storage. Therefore, the glass transition temperature and viscosity are crucial parameters. These parameters allow for predicting the correlation between the resin structure and the properties of the powder coatings.

If the resin has low viscosity and T*_g_*, the powder may stick together, making it impossible to apply the powder coating using an electrostatic gun. Conversely, if the viscosity and T*_g_* of the acrylic resin are too high, problems may arise with the melting and flow of powder coatings, resulting in poor final product properties. Table 4 presents the T*_g_* values and viscosity of acrylic resins. The recommended T*_g_* of powder coatings is approximately 50 °C [46]. The resin HEMA/6MMA/0.5*n*BA/0.5DA exhibited the most optimal T*_g_* value (T*_g_* = 50.31 °C), which positively influenced the final properties of the powder coatings. Additionally, the viscosity of this resin had a beneficial effect on the flow and leveling properties of the powder coatings, as surfaces based on this resin did not exhibit defects such as orange peel or craters. However, monomers such as *t*BA or BAZ had an adverse effect on the formation of powder coatings.

The chemical structure of the synthesized acrylic resins was confirmed using ^1^H-NMR spectra (Figure 1).

The signal in the range of 0.60–1.00 ppm (assigned as ‘A’) is characteristic for the hydrogen atoms of the methyl groups derived from the main polymer chain [47]. In this region it is also visible that the signal comes from the protons of the methyl group at the end of the dodecyl and butyl substituent marked as ‘F’. In the range of 3.60 ppm, the protons of the methyl groups derived from MMA (–OCH_3_) are observed (assigned as ‘B’). The methylene groups adjacent to oxygen (–OCH_2_-) assigned at ‘D’, ‘G’ and ‘H’ derived from HEMA, BA and DA are visible in the range of 3.80–4.10 ppm [48]. The hydrogen atom (‘I’) derived from the hydroxyl group is seen at 3.15 ppm. Signals in the range of 1.10–1.70 ppm come from protons of aliphatic methylene groups assigned as ‘E’ and ‘E’’ of BA and DA. Signals originating from protons of methylene groups forming the main polymer chain (assigned as ‘C’) are visible at 1.70–1.90 ppm [47,49]. A common problem with acrylic resins synthesized in bulk polymerization is the presence of small amounts of free monomers. In the spectrum ^1^H NMR (Figure 1), a small number of unreacted monomers can be seen in the range of 5.60–6.40 ppm. In order to remove free monomers, the acrylate resin was heating at 80 °C by 1 h in oven.

### 4.3. The Crosslinking Process and Properties of Powder Coatings

After obtaining acrylic resin, the next step was the selection of an appropriate crosslinking agent. Due to the presence of hydroxyl groups in the resin, a blocked polyisocyanate named Vestanat^®^ B 1358/100 by Evonic was suitable. To assess the course of the crosslinking reaction, a DSC analysis was conducted for the uncured powder coatings. To assess the course of the crosslinking reaction, a DSC analysis was conducted for the uncured powder coating compositions (Figure 2).

In order to verify the complete crosslinking, two heating cycles were conducted. In the range of 45–65 °C, the glass transition of the powder coating was observed. During the first heating cycle, in the range of 110–160 °C, a noticeable broad exothermic peak suggests the progress of the powder coating curing process. To confirm the complete curing process of the coatings, a second heating cycle was performed. The absence of a signal in the range of 110–160 °C during the second heating cycle indicates the complete occurrence of this process in the first cycle. This temperature range confirms the attainment of a low-temperature curing powder coating (160 °C for 15 min). The crosslinking process of other samples was very similar, which indicates that the influence of the chemical composition of the resin had no significant impact on this process. Additionally, the crosslinking of the powder coatings was confirmed by using the polymerization test performed according to Qualicoat [21]. After taking the test, no noticeable changes were observed. Moreover, the samples, after the curing stage, exhibited no defects such as orange peel or craters. Subsequently, the powder coatings underwent further tests, the results of which are provided in Table 5.

The thickness of the obtained powder coatings ranged from 60 to 80 µm, meeting the Qualicoat requirements [21].

The study on the influence of chemical structure on the properties of powder coatings focused on the monomers such as *n*BA, *t*BA, EA, EMA, BAZ and DA. HEMA and MMA monomers were present in all samples in the same amount. HEMA was responsible for the crosslinking process and the MMA contributed to the stability of composition. The flow properties of uncured powder compositions were examined. Samples L_HEMA/6MMA/*t*BA, L_HEMA/6MMA/EA and L_HEMA/6MMA/EMA showed very poor flowability. This reduced flowability led to an increased roughness and decreased gloss of the cured coatings. Additionally, these coatings exhibited weak properties such as adhesion and scratch resistance compared to other powder coatings. The presence of tert-butyl substituents into *t*BA increase roughness and hardness but decreases flexibility causing a problem with brittleness of coatings compared with *n*BA, which contains a linear substituent with the same number of carbon atoms. A similar stiffening effect was observed when comparing the effects of EA and *n*BA (samples L_HEMA/6MMA/EA and L_HEMA/6MMA/*n*BA). The shorter aliphatic chain ethyl substituent on the ester group in EA increased roughness while reducing gloss, scratch resistance and adhesion to the steel. Comparing the effects of EA and EMA, the stiffening effect of the presence of a methyl group in the α-position of the vinyl bond in EMA (sample L_HEMA/6MMA/EA and L_HEMA/6MMA/EMA) was evident, causing worse flowability and gloss of the coating. Therefore, the presence of a methyl group in the α-position of the vinyl bond is important in designing powder coatings based on acrylic resins. To compare the effect of a substituent in the ester group with aliphatic and aromatic structure, BAZ was used. Coatings based on BAZ exhibited good flowability, the highest relative hardness, cupping and were classified as “semi-gloss” coatings. While styrene is a common monomer used in commercial coatings due to its cost-effectiveness and enhancement of mechanical properties, it suffers from brittleness and lack of flexibility [50].

The introduction of DA into the resin was aimed at addressing the flexibility issue associated with typical acrylic resins. However, the long chain attached to the ester group significantly improved flowability and cupping but decreased to low the glass transition temperature (T*_g_*), which led to a problem with the storage and application of powder coatings. Therefore, a resin containing *n*BA and DA (HEMA/6MMA/0.5*n*BA/0.5DA) was developed. The coating based on this resin exhibited favorable properties including good flowability, gloss, adhesion and relative hardness. Additionally, it demonstrated the highest contact angle (93.53 degrees) and scratch resistance (550 g), excellent cupping (13.38 mm) and improved impact resistance compared to other samples. To further understand this behavior, crosslinking density and thermal analysis were performed.

The crosslinking density (*v_e_*) was determined using dynamic mechanical analysis (DMA). This technique measures the mechanical forces applied in a function of temperature in a polymer. The dependence of storage modulus E’ on temperature for sample L_HEMA/6MMA/0.5*n*BA/0.5DA is presented in Figure 3.

The modulus in the rubbery plateau (*E’_min_*) was determined based on this dependence. Crosslinking density *v_e_* as the number of moles of effective network chains per volume of 1 m^3^ (mol/m^3^) was determined using equation (1). In this equation, R is the gas constant (8.314 J/mol K) and T is the temperature (K) at which *E’_min_* was determined.
(1)$$v_{e}=\frac{E’}{3RT}$$

The crosslinking density of the L_HEMA/6MMA/0.5*n*BA/0.5DA coating was determined to be *v_e_* =111.2 mol/m^3^, with values reported in the literature for powder coatings [40]. Furthermore, the L_HEMA/6MMA/0.5*n*BA/0.5DA coating exhibited a high storage modulus, confirming its excellent mechanical properties. It is known that higher storage modulus and higher T*_g_* are indicative of better properties.

Additionally, the thermal stability of these powder coatings was assessed via thermogravimetric analysis (TGA). The coating L_HEMA/6MMA/0.5*n*BA/0.5DA displayed decomposition in three stages in the temperature range of 160–470 °C, as illustrated in Figure 4.

The first step occurs at the temperature of the maximum mass loss rate of T_max1_ = 213 °C and is related to the deblocking of polyisocyanate. The second step occurs at the temperature of the maximum mass loss rate of T_max2_ = 293 °C and is related to the scission of the urethane bond, while the third step (T_max3_ = 404 °C) is associated with the degradation of the acrylic resin segments. The mass losses at the first, second and third stages of degradation amount to 8.9%, 19.1% and 70.7%, respectively. The temperature of 5% mass loss is 213 °C. On Figure 5 is shown thermograms of the L_HEMA/6MMA/*n*BA sample.

In the comparison of thermograms of the L_HEMA/6MMA/0.5*n*BA/0.5DA sample, no significant difference is visible. The degradation process of the powder coatings initiates with the breakdown of the weakest bonds, namely the urethane bonds formed between the polyisocyanate and the resin. This underscores the importance of understanding the bond strengths within the polymer matrix, as it dictates the onset and progression of degradation during thermal treatment.

## 5. Conclusions

The study focused on investigating the impact of acrylic monomer chemical structure on the properties of powder coatings formulated with polyacrylate resin. Gel permeation chromatography (GPC) revealed an average molecular weight of approximately 7000 Da for the resins, a typical value for acrylic resins used in powder coatings. Parameters such as glass transition temperature (T*_g_*) and viscosity were determined for the polyacrylate resins, both of which were influenced by the resin’s chemical structure. These parameters play crucial roles in the formulation, storage, application and final properties of powder coatings. Understanding the correlation between the chemical structure, viscosity and T*_g_* of the resin provided valuable insights for optimizing powder coating formulations.

Dynamic scanning calorimetry (DSC) was employed not only to determine the T*_g_* of the resins but also to examine the crosslinking process. The coatings underwent crosslinking at 160 °C, enabling low-temperature curing and allowing for application on heat-sensitive materials. This low-temperature curing capability broadens the range of substrates onto which powder coatings can be applied, enhancing their versatility and utility in various industries.

The powder coatings based on the resins containing *t*BA, EA and EMA were characterized by high roughness and poor physical–mechanical properties. The best properties were noted for coating L_HEMA/6MMA/0.5*n*BA/0.5DA, which was characterized by good impact resistance, scratch resistance (550 g) and adhesion to the steel surface, a high water contact angle (93.53 deg.) and excellent cupping (13.38 mm). These results highlight the effectiveness of the resin formulation in achieving desirable properties, making it a promising candidate for various coating applications.

This powder coating is also characterized by high crosslinking density (111.2 mol/m^3^) and good thermal stability.

Indeed, enhancing the elasticity of acrylic resins while maintaining other essential properties expands their potential applications, particularly in coating structures with irregular shapes or sharp edges. In such cases, excessively rigid coatings are prone to cracking, compromising both the aesthetic appearance and protective function of the coating. By increasing elasticity, acrylic resins can better accommodate substrate movements and deformations, thereby minimizing the risk of cracking and ensuring the long-term durability of the coating. This advancement allows for broader utilization of acrylic-based powder coatings in diverse industries, including those requiring coatings for challenging surface geometries and environmental conditions.

## Figures and Tables

**Figure 1 materials-17-01655-f001:**
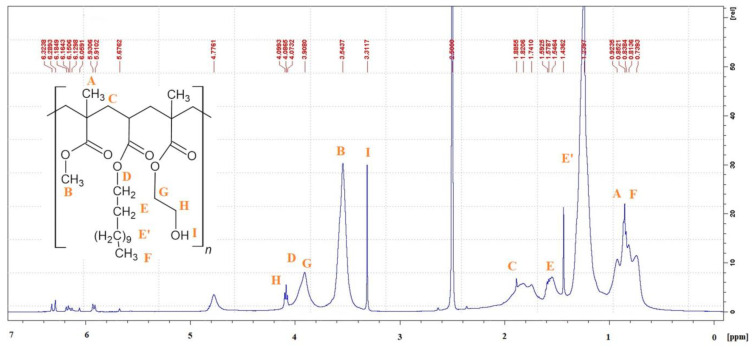
^1^H-NMR spectrum of HEMA/6MMA/DA resin.

**Figure 2 materials-17-01655-f002:**
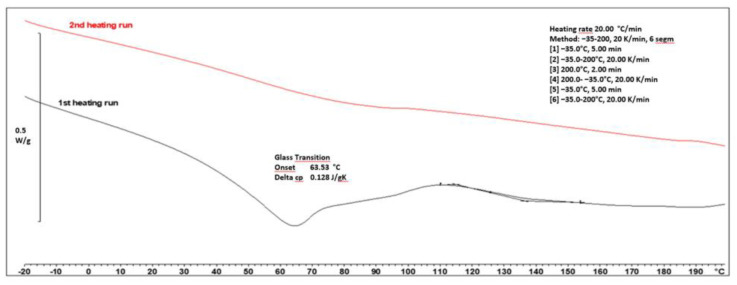
The DSC curves of L_HEMA/6MMA/0.5*n*BA/0.5DA.

**Figure 3 materials-17-01655-f003:**
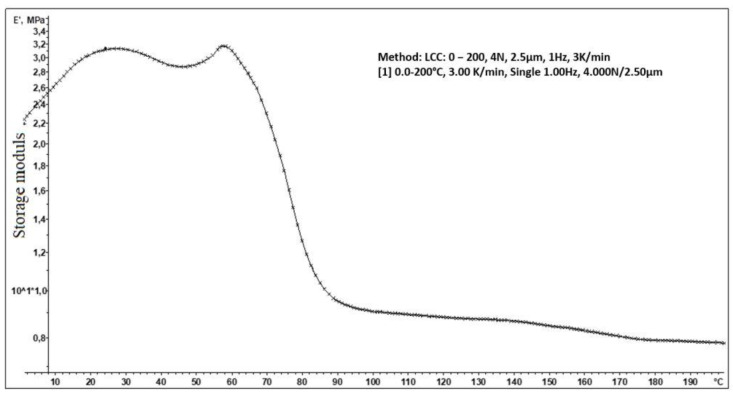
DMA curves of L_HEMA/6MMA/0.5*n*BA/0.5DA.

**Figure 4 materials-17-01655-f004:**
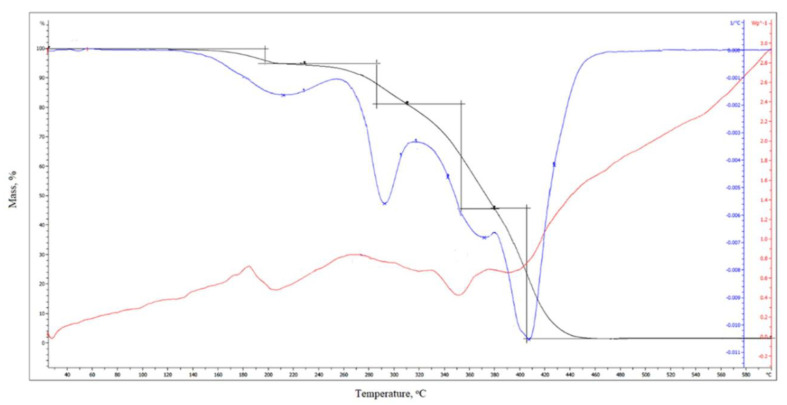
TG, DTG and DTA thermograms of L_HEMA/6MMA/0.5*n*BA/0.5DA sample.

**Figure 5 materials-17-01655-f005:**
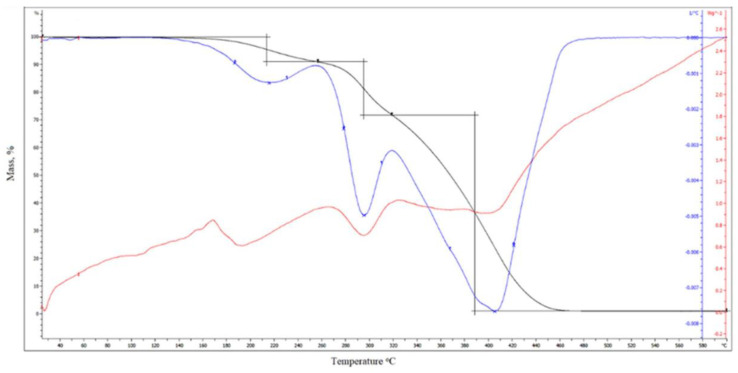
TG, DTG and DTA thermograms of L_HEMA/6MMA/*n*BA sample.

**Table 1 materials-17-01655-t001:** Qualitative and quantitative composition of the synthesized acrylic resins.

Resin Symbol	2-hydroxyethyl methacrylate (HEMA)	methyl methacrylate (MMA)	n-butyl acrylate (*n*BA)	tert-butyl acrylate (*t*BA)	ethyl acrylate (EA)	ethyl methacrylate (EMA)	benzyl acrylate (BAZ)	dodecyl acrylate (DA)
HEMA/6MMA/*n*BA	1	6	1	-	-	-	-	-
HEMA/6MMA/*t*BA	1	6	-	1	-	-	-	-
HEMA/6MMA/EA	1	6	-	-	1	-	-	-
HEMA/6MMA/EMA	1	6	-	-	-	1	-	-
HEMA/6MMA/BAZ	1	6	-	-	-	-	1	-
HEMA/6MMA/DA	1	6	-	-	-	-	-	1
HEMA/6MMA/0.5*n*BA/0.5DA	1	6	0.5	-	-	-	-	0.5

**Table 2 materials-17-01655-t002:** The summarizing of different types of (meth)acrylate monomers used and their influence on resin properties.

Name of Monomer	Symbol	Chemical Structure	Resin Properties	Reference
methyl methacrylate	MMA	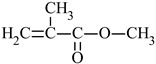	-good thermal stability-high Young’s Modulus-low elongation at break-high scratch resistance-good chemical resistance	[32,33]
*n*-butyl acrylate	*n*BA	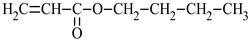	-high flexibility-low thermal stability	[34]
*tert*-butyl acrylate	*t*BA	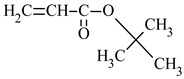	-improves impact resistance-more thermally stable than *n*PBA-hydrophobic properties	[35,36]
ethyl acrylate	EA	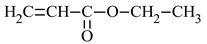	-hydrophobic properties	[37]
ethyl methacrylate	EMA	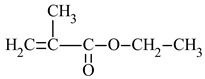	-better resistance to weather conditions than PEA	[38]
benzyl acrylate	BAZ	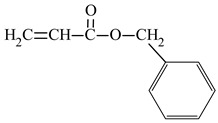	-increasing strength and hardness-decreasing adhesive properties	[39]
dodecyl acrylate	DA	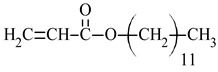	-high hydrophobicity-high flexibility	[40,41]
2-hydroxyethyl methacrylate	HEMA	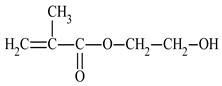	-hydrophilic properties-improves mechanical properties (e.g., flexibility)	[42]

**Table 3 materials-17-01655-t003:** Average molecular masses of synthesized acrylic resins.

	Mass Unit	Average Molecular Mass Number (*M*_n_)	Average Molecular Mass Weight (*M*_w_)	Molecular Mass Z-Average (*M*_z_)
Resin Symbol		[Da]	[Da]	[Da]
HEMA/6MMA/*n*BA		8860	53,610	22,760
HEMA/6MMA/*t*BA		9820	55,500	18,520
HEMA/6MMA/EA		8240	34,930	15,030
HEMA/6MMA/EMA		7420	17,360	36,270
HEMA/6MMA/BAZ		7540	13,910	21,180
HEMA/6MMA/DA		6670	39,410	23,550
HEMA/6MMA/0.5*n*BA/0.5DA		8020	53,320	21,700

**Table 4 materials-17-01655-t004:** Glass transition temperature (T*_g_*) and viscosity of obtained acrylic resins.

Resin Symbol	Glass Transition Temperature (T*_g_*) [°C]	Viscosity [Pa*s]
HEMA/6MMA/*n*BA	54.03	23.85
HEMA/6MMA/*t*BA	78.29	22.20
HEMA/6MMA/EA	54.19	19.00
HEMA/6MMA/EMA	52.93	24.30
HEMA/6MMA/BAZ	42.62	36.15
HEMA/6MMA/DA	39.40	17.25
HEMA/6MMA/0.5*n*BA/0.5DA	50.31	20.73

**Table 5 materials-17-01655-t005:** Summary of physical and mechanical parameters of the powder coatings.

Physical or Mechanical Parameter	Powder Coating Symbol						
	L_HEMA/6MMA/*n*BA	L_HEMA/6MMA/*t*BA	L_HEMA/6MMA/EA	L_HEMA/6MMA/EMA	L_HEMA/6MMA/BAZ	L_HEMA/6MMA/DA	L_HEMA/6MMA/0.5*n*BA.0.5DA
Flowability [cm]	1.30	0.90	0.95	0.75	2.20	5.00	4.40
Roughness: *Ra* *Rz*	0.69/2.74	1.79/8.73	2.88/14.12	6.33/11.73	1.42/6.79	1.09/5.11	0.42/2.05
Gloss 60 °C [GU]	83.45	34.73	34.24	11.73	63.12	64.62	79.63
Adhesion to the steel substrate [0—good 5—bad]	0	0	1	1	1	0	0
Relative hardness [-]	0.54	0.56	0.55	0.51	0.67	0.33	0.50
Scratch resistance [g]	500	450	250	250	550	300	550
Contact angle [deg]	85.40	83.93	87.91	83.91	87.02	93.14	93.53
Impact resistance [J/cm^2^]	15	10	15	10	20	25	30
Cupping [mm]	5.43	4.37	5.12	5.07	9.96	11.18	13.38

## Data Availability

The original data presented in the study are included in the article. Additional raw data supporting the conclusions of this article will be made available by the authors on request.

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
