# Peer review of "Correlation between the Chemical Structure of (Meth)Acrylic Monomers and the Properties of Powder Clear Coatings Based on the Polyacrylate Resins"

_materials, 2024, doi:10.3390/ma17071655_

Round 1

Reviewer 1 Report

Comments and Suggestions for Authors

Summary

            Due to requirements from the EU as well as individual customers, researchers have sought to create new, nontoxic products for the paints and coatings market. Numerous poly(meth)acrylate 

resins were made using HEMA, MMA, nBA, tBA, DA, EA, and BAZ; all the resins were made with 1:6 HEMA:MMA (molar equivalents) and one molar equivalent of the other monomers. One of the resin mixtures consisted of half molar equivalents of nBA and DA in addition to the HEMA and MMA. This resin, based on its desirable physical characteristics, was converted to a powder coating via crosslinking. This powder coating had a scratch resistance of 550 g, good steel adhesion, 93.53˚ contact angle, cupping of 13.38 mm, and a crosslinking density of 111.2 mol/m3.

General Writing Comments

            Sections 4.2 and 4.3 of the results and discussion section would fit better in the introduction. A lot of information in the first paragraph of the conclusion is information that was already given or should be kept in the introduction. The writing needs many improvements in optimized structure and language. Table 1 has some problems too, where some resin symbols do not match up with the numbers on the side. 

General Conceptual Comments

            It seems like all the methodology is standard and typical in the polymers field and acceptable to be used for this study. The TGA data is logical, considering the number of different organic components, besides the parts where the mass % increases. It could be useful to try even more combinations of acrylate monomers to see if an even better resins can be made; perhaps this has been done already. 

Comments on the Quality of English Language

English language needs extensive revision and should be addressed before final publication

Author Response

We would like to thank you for the comments. We have analyzed all the comments and I would like to submit the following answers:

Review Report (Round 1)

General Writing Comments

Sections 4.2 and 4.3 of the results and discussion section would fit better in the introduction. A lot of information in the first paragraph of the conclusion is information that was already given or should be kept in the introduction. The writing needs many improvements in optimized structure and language.

ANSWER:

Thank you for this remark. Sections 4.2 and 4.3 have been revised according to suggestion, and marked in red.

Table 1 has some problems too, where some resin symbols do not match up with the numbers on the side. 

ANSWER:

Thank you for this remark. Table 1 has been corrected and marked in red.

General Conceptual Comments

It seems like all the methodology is standard and typical in the polymers field and acceptable to be used for this study.

The TGA data is logical, considering the number of different organic components, besides the parts where the mass % increases.

ANSWER:

Thank you for this remark. During TGA degradation, increase mass loss %, not mass%. It has been corrected and marked in red.

It could be useful to try even more combinations of acrylate monomers to see if an even better resins can be made; perhaps this has been done already. 

ANSWER:

Thank you for this remark. The more combinations of acrylate monomers have been made and the best results have been selected.

English language needs extensive revision and should be addressed before final publication

ANSWER:

Thank you for this remark. Linguistic correction has been applied and marked in green in the text.

Reviewer 2 Report

Comments and Suggestions for Authors

Please, see attached file.

Comments on the Quality of English Language

Dear Editor,

as mentioned in my attached file for the auhtors, the quality of English laguage should be drastically improved.

Best Regards

Georgios Bokias

Author Response

We would like to thank you for the comments. We have analyzed all the comments and I would like to submit the following answers:

Review Report (Round 2)

The present work tries to correlate the structure of acrylic monomers with the properties of powder coatings based on polyacrylate resins. The main purpose is to formulate the chemical structure of the resin, aiming at a powder coating with the desired technical characteristics for this application.

 The manuscript could be accepted for publication. However, it suffers from major problems:

  1. a) The major drawback of the manuscript is the use of English language. I feel that it is necessary the authors to ask the help of a native English-speaker scientist, at least.

ANSWER:

Thank you for this remark. Linguistic correction has been applied and marked in green in the text.

  1. b) Figure 1 is useless and It can be clearly described in just two lines.

ANSWER:

Figure 1 has been removed.

  1. c) Section 4.1 was clearly introduced in Introduction. No additional information is given, here. It could be removed. Figure 2, if necessary, can be discussed as introductory to the next section.

ANSWER:

Thank you for this remark. Section 4.1 and Figure 2 have been removed.

  1. d) Section 4.2. Here, a too lengthy presentation (from line 331 up to line 375) of the properties of the resins is made. Most of the properties discussed are evident or common knowledge. Most important properties are already summarized in Table. My suggestion is that this pdiscussion should be much shorter (10-20 lines) and concise, emphasizing just the main reasons for these choices.

ANSWER:

Thank you for this remark. 4.2 and 4.3 have been modified and shortened, and marked in red

  1. e) Line 171: “… containing 5 mmol/l..”. Of what?

ANSWER:

Thank you for this remark. All samples were dissolved in tetrahydrofuran (HPLC grade) containing 5 mmol/l of the resin at a temperature of 22 °C. Correction has been applied in the text and marked in red.

  1. f) Table 2: Please, check the chemical structure of HEMA, the OH groups is partly “hidden”.

ANSWER:

Thank you for this remark. The chemical structure of HEMA has been corrected.

  1. g) Section 4.3. A lengthy presentation for literature is made, here. However, it nis unclear whether there are findings really crucial for the present work and the present results. In any case, this paragraph is not related to any results of the present study.

ANSWER:

Thank you for this remark. Section 4.3 have been modified and shortened, and marked in red

The literature part has been removed.

  1. h) Table 3. Typically for free radical polymerizations, the polydispersity indexes (Mw/Mn) are between 1.5 and 2, and usually around 2. However, for some samples polydispersity indexes more than 5 are reported. Is there any explanation?

ANSWER:

In this work, bulk free radical polymerization was employed because the obtained product can be directly utilized for powder coating production. The high reactivity of acrylic monomers and the associated rapid heat release can lead to localized overheating, especially during mass polymerization where heat dissipation is more challenging, resulting in greater polydispersity. Additionally, a small amount of unreacted monomers always remains in the high viscosity resin being formed, contributing to increased polydispersity. In the case of solvent-based methods, the degree of polydispersity will be significantly lower because heat dissipation is easier, and unreacted monomers can be removed along with the solvent. However, to use the product in powder coating, the solvent must be thoroughly removed, which entails additional costs and VOC emissions.

  1. i) Figure 3. The attribution of peaks should be rechecked carefully. For example, though a reference is given, peak A is usually found at 1ppm or lower, peak B at around 3.5 ppm, peaks G and H (if correctly attributed should have the same area), peak F is wrongly attributed in the spectrum (whlile correctly discussed in lines 437-438), etc.

ANSWER:

Thank you for this remark. Figure 3 (Figure 1 after revision) has been corrected.

  1. j) NMR analysis. Is it possible to determine the actual composition of the final products through the NMR analysis of the sample?

ANSWER:

Determining the actual composition of the final product through 1H-NMR is possible when at least one of the signals originating from each monomer does not overlap with others. This is most often achievable in the case of a copolymer composed of monomers whose signals do not overlap, such as from the proton of the HEMA methyl group and from the proton of the methyl group in the substituent adjacent to the ester group of BA. In such cases, the composition can be determined based on the integration of signals originating from the protons of individual monomers.

Unfortunately, in this case, it is not possible to accurately determine the actual composition of acrylic resin based on the 1H-NMR spectrum. This is due to the overlapping of protons from the methyl group -CH3 originating from HEMA and MMA monomers, as well as protons from the aliphatic chain of BA and DA.

Reviewer 3 Report

Comments and Suggestions for Authors

This is an interesting and practical research. I just have one concern, could you provide more mechanistic analysis on the optimal performance of the HEMA/6MMA/0.5BA/0.5DA sample?

Comments on the Quality of English Language

Too many typos even in the abstract, such as "the DSC was also used to check the course."

Author Response

We would like to thank you for the comments. We have analyzed all the comments and I would like to submit the following answers:

Review Report (Round 3)

This is an interesting and practical research. I just have one concern, could you provide more mechanistic analysis on the optimal performance of the HEMA/6MMA/0.5BA/0.5DA sample?

ANSWER:

In order to confirm the optimal performance of the HEMA/6MMA/0.5BA/0.5DA sample, an additional impact resistance test was performed. This test additionally confirmed the best resistance of this sample to sudden impact forces.

Too many typos even in the abstract, such as "the DSC was also used to check the course."

ANSWER:

The typos even in all text has been corrected and marked in red.

Round 2

Reviewer 2 Report

Comments and Suggestions for Authors

The authors have followed all suggestions and the manuscript is substantially improved, in terms of quality of presentation and clarity.

A minor (important) comment:

The attribution of peaks in Figure 1  is still not correct. Namely, the large peak (where now methylene  groups C are assigned) probably should be attributed to the repetitive CH2 groups of DA. Moreover, in my opinion, peak A should be assigned together with peak F (however, this should be checked by the authors).

Author Response

A minor (important) comment:

The attribution of peaks in Figure 1  is still not correct. Namely, the large peak (where now methylene  groups C are assigned) probably should be attributed to the repetitive CH2 groups of DA. Moreover, in my opinion, peak A should be assigned together with peak F (however, this should be checked by the authors).

ANSWER:

Thank you for this remark. You're right. We checked the attribution of peaks in Figure 1 again:

The signal in the range of at 0.6-1.0 ppm (assigned as ‘A’) is characteristic for the hydrogen atoms of the methyl groups derived from the main polymer chain [36]. In this region is also visible the signal comes from the protons of the methyl group at the end of the dodecyl and butyl substituent marked as ‘F’. Signals in the range of 1.10-1.70 ppm come from protons of aliphatic methylene groups assigned as ‘E’ and ‘E’’ of BA and DA. Signals originating from protons of methylene groups forming the main polymer chain (assigned as ‘C’) are visible at 1.7-1.9ppm according to the publication [36, 38] and the sdbs database[https://sdbs.db.aist.go.jp/sdbs/cgi-bin/direct_frame_top.cgi.

Figure 1 ha been corrected, Section 4.2 has been revised and changes are highlighted in yellow.

[36]     Massoumi B., Jaymand M.; Chemical and electrochemical grafting of polythiophene onto poly(methyl methacrylate), and its electrospun nanofibers with gelatin Ramis, Journal of Materials Science: Materials in Electron 2016 12, 27. https: //doi.org/10.1007/s10854-016-5413-5.

[38] Babac C., Utkan G., David G., Simionescu B., Piskin E., Production of nanoparticles of methyl methacrylate and butyl methacrylate copolymers by microemulsion polymerization in the presence of maleic acid terminated poly( N-acetylethylenimine) macromonomers as cosurfactant, Europen Polymer Journal 2004 40, 1947-1952. https: //doi.org/10.1016/j.eurpolymj.2004.03.004